# Optogenetic stimulation of single ganglion cells in the living primate fovea

Peter J Murphy[1,2]*, Juliette E McGregor[2,3], Zhengyang Xu[1,2], Qiang Yang[2,3], William Merigan[1,3], David R Williams[1,2]

[1]The Institute of Optics, University of Rochester, Rochester, United States; [2]Center for Visual Science, University of Rochester, Rochester, United States; [3]Flaum Eye Institute, University of Rochester, Rochester, United States

## eLife Assessment

This article shows that it is possible to optogenetically activate single retinal ganglion cells in vivo in monkeys. This is an **important** step toward causal tests of the role of specific ganglion cell types in visual perception. The article presents **convincing** evidence for the promise of the approach, but further work will be needed to fully explore its limitations and specificity.

**Abstract** Though the responses of the rich variety of retinal ganglion cells (RGCs) reflect the totality of visual processing in the retina and provide the sole conduit for those processed responses to the brain, we have much to learn about how the brain uses these signals to guide behavior. An impediment to developing a comprehensive understanding of the role of retinal circuits in behavior is the paucity of causal studies in the intact primate visual system. Here, we demonstrate the ability to optogenetically activate individual RGCs with flashes of light focused on single RGC somas in vivo, with no evidence for activation of neighboring cells. The ability to selectively activate specific cells is the first step toward causal experiments that directly link retinal circuits to visual experience and behavior.

*For correspondence:
pmurp16@ur.rochester.edu

## Introduction

Though only a few classes of light-sensitive cells capture the retinal image in the primate eye, evolution has produced many more classes of retinal ganglion cell (RGC), on the order of 20, each morphologically and physiologically distinct and independently tiling the retina, to convey visual information to the brain. This dramatic expansion of cell classes in the retinal output, combined with microelectrode recordings in excised tissue, has made it increasingly clear that the diversity of RGCs reflects instead a surprising amount of complex early processing in the mammalian retina, processing that had previously been thought to reside deeper in the brain (*Gollisch and Meister, 2010*). There is evidence, for example, for specialized circuits that convey directional selectivity (DS) (*Barlow et al., 1964*), segregate object from background motion (*Lettvin et al., 1959*), and anticipate moving stimuli (*Berry et al., 1999*). However, we remain uncertain about the direct functional role of these RGC classes because, while ex vivo recordings can characterize the specific stimulus requirements of RGC classes, they are often made in the excised retina that is detached from the downstream visual pathways they serve. Such studies preclude direct manipulations of RGC activity to observe the impact on visual experience or behavior in the awake primate. For example, we cannot say with certainty whether the DS cells recently observed in the non-human primate (NHP) are involved in eye movements, the perceptual experience of motion, or both. Indeed, there is even lingering controversy

about the perceptual roles of the most common primate RGCs, the midget and parasol ganglion cells (*Freedland and Rieke, 2022*; *Patterson et al., 2019*).

To address these controversies, one exciting possibility for the future is the development of studies in which the effect of manipulating RGC responses in the awake behaving animal can be measured with psychophysical or behavioral measures. There have been formidable challenges to deploying this causal paradigm to study retinal circuits because of their inaccessibility inside the moving eye of the awake behaving primate. Advances in high-resolution retinal imaging with adaptive optics (AO) now make it possible to observe single RGCs in the living eye (*Gray et al., 2008*). AO has also enabled very accurate eye tracking, making it possible to not only image but also stimulate single photoreceptors repeatedly with focused flashes of light (*Schmidt et al., 2018a*; *Schmidt et al., 2018b*; *Tuten et al., 2017*) in the moving human eye. In addition, viral vectors now allow expression of calcium indicators such as GCaMP6s (*Yin et al., 2013*; *Godat et al., 2022*), enabling recording from single RGCs in vivo. Finally, optogenetic activation of RGCs has been demonstrated in both macaque (*McGregor et al., 2020*) and human (*Sahel et al., 2021*) retina in vivo. These recent technical advances taken together now offer the possibility of causal experiments in which single RGCs are optogenetically stimulated in the living primate eye while the behavioral consequences are measured. Here, we demonstrate the first step toward establishing this capability: showing successful stimulation of single RGCs in the living primate eye.

## Results

RGC activity was modulated via the optogenetic ChrimsonR and monitored via fluorescence imaging of the calcium indicator GCaMP6s. Isolated foveal RGCs were chosen for individual stimulation, being targeted with short pulses of light intended to elicit activity in that cell only. The response of both the selected cell and all nearby cells was monitored to determine activation status. Chosen cells and their fluorescence traces can be seen in *Figure 1*. Targeted cells showed large responses to the optogenetic probe. The GCaMP responses rise to peak quickly and then decay over several seconds. This time course is consistent with GCaMP responses to cone excitation of RGCs measured via in vivo AOSLO imaging (). *Figure 2* shows all cells' $\Delta$ F/F values plotted against their distance from the targeted cell. Non-targeted cells showed no significant response above the background noise in contrast to the targeted cells, the responses of which were between three and four times the background noise floor. While we cannot exclude the possibility that surrounding cells responded to the flash below the noise threshold of our measurements, clearly these conditions strongly favor the excitation of the target cell over that of the neighbors we were able to record from. We cannot rule out the possibility that, due to the random nature of the vector expression process, some of these non-targeted cells were inactive because they failed to express ChrimsonR. The degree of GCaMP6s and ChrimsonR expression varies from one RGC to another, introducing the possibility that some cells may only have significant expression of one or neither agent. This means that some optogenetically active cells may have had no or weak GCaMP6s expression, diminishing our ability to observe their responses in our recordings. Despite this uncertainty, we can be reasonably sure that most cells are strongly co-expressing both agents. This is achieved via fluorescence imaging of the tdTomato marking ChrimsonR expression, which shows substantial overlap with the observed GCaMP6s fluorescence.

## Discussion

Though this method has limitations, these may be overcome via further refinements of the technique. For this initial demonstration, we deliberately selected the most favorable conditions, especially by selecting cells to target that were spatially isolated from neighboring cells. Here, we show results for cells at the inner edge of the ring of RGCs serving the primate fovea. The ability to extrapolate the method to cells closely packed on all sides by neighboring cells, as occurs at slightly larger eccentricities, between 0.5 and 2.5°, deeper in the foveal ring of labeled cells, has yet to be explored. In this region, RGCs have little to no separation between cell somas. The transverse spatial resolution of an AOSLO is roughly 1.6 µm, which is more than adequate for targeting the much larger soma diameter. However, the axial resolution of an AOSLO is roughly 25 µm, roughly twice the soma diameter. This means that there would be some optogenetic excitation of the cells immediately above and below the cell of interest. But even in this less favorable case, the balance of excitation could

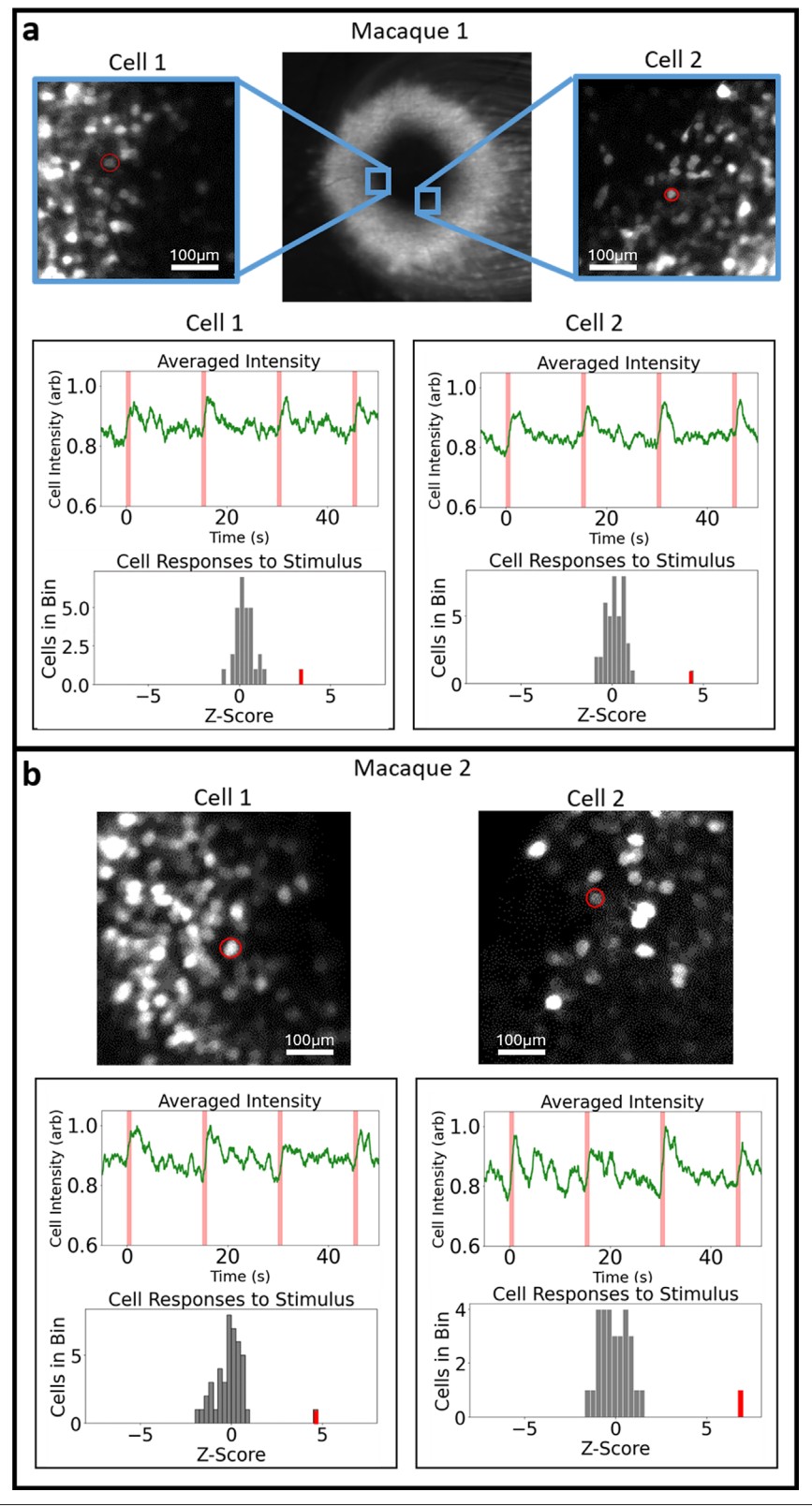

**Figure 1.** Cells chosen for stimulation. The cell circled in red marks the targeted cell. The averaged fluorescence traces for each cell show the integrated intensity over the cell's soma. The red marks on the time axis represent the onset of the 800 ms optogenetic stimulus. The histograms show each cell's response in terms of z score. (**a**) Results from male macaque. (**b**) Results from female macaque. All targeted cells are at similar eccentricities (1.5°).

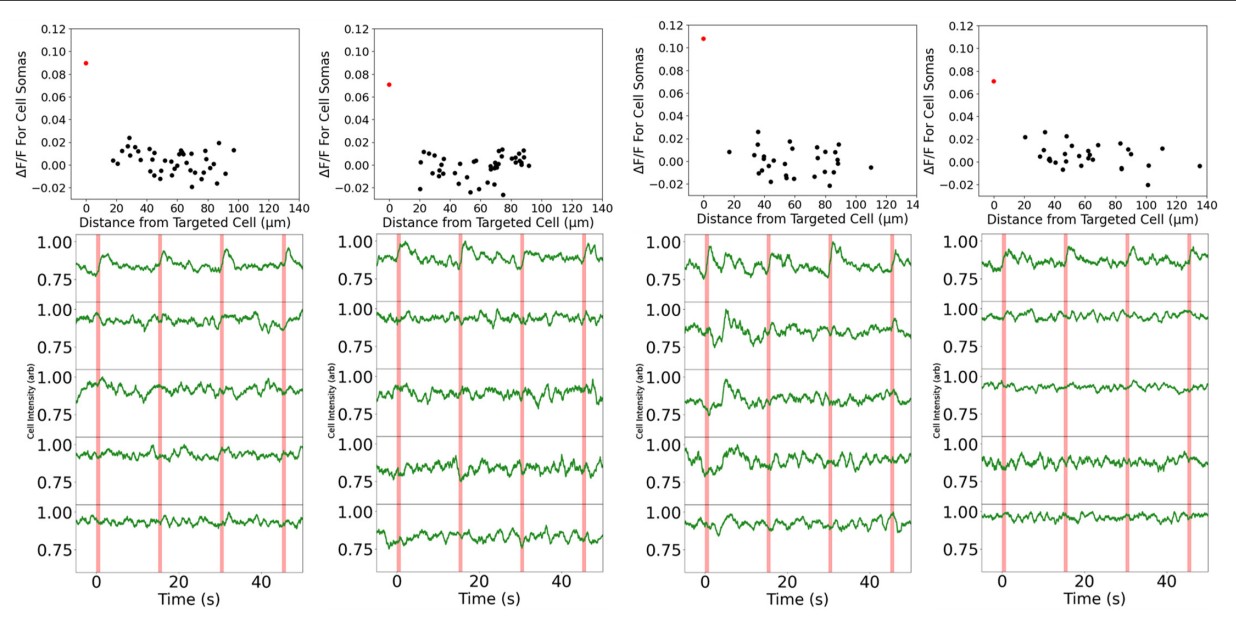

**Figure 2.** ΔF/F for each cell soma plotted against that cell's distance from the targeted cell. Note that even the nearby (<50 µm) cell somas do not show a significantly elevated response (p>>0.05, unpaired *t*-test) than other cells at more distant locations. The leftmost point on each plot, colored in red, corresponds to the targeted cell itself. The traces for the four closest neighbors are shown below each scatter plot.

be shifted strongly to a subset of RGCs, even if perfect isolation were not possible. The ability to isolate single cells becomes far more favorable again in the periphery, where the RGC somas are in a monolayer. However, in this retinal region, the RGC somas and their cone receptive fields lie on top of each other, precluding the use of the spatial offset between cone receptive fields and their somas to separate cone and RGC optogenetic excitation. The use of two-photon excitation coupled with the use of a more sensitive optogenetic, such as ChRmine, may mitigate this concern by increasing the optogenetic excitation while reducing the efficacy of light falling on photoreceptors (*Marshel et al., 2019*). Other approaches to increasing the selective activation of targeted cells may include cortically-targeted retrograde injections or class-specific viral vectors.

The demonstration of in vivo selective activation of individual RGCs lays the foundation for future causal studies of RGC function. The causal experimental approach has been successful in understanding the roles of neuronal classes in the cortex of NHPs for many years, such as the work of Newsome and colleagues who showed that electrical activation of DS cells in MT modifies the animal's choices in motion discrimination tasks (*Salzman et al., 1992*; *Cicmil and Krug, 2015*; *Parker and Newsome, 1998*). More recently, this paradigm has incorporated optogenetic activation to investigate the function role of localized neural circuitry in cortex (*Jazayeri et al., 2012*; *Soma et al., 2019*). Applied to RGCs, this approach would involve first, in vivo calcium imaging to classify each RGC, followed by psychophysical experiments combined with optogenetic stimulation designed to explore each cell's role in visual behavior. Such experiments would establish a direct link between activation of an individual RGC and visual behavior and could help determine whether a given cell class contributes to perceptual experience, subconscious visual behavior, or both. These experiments will face additional challenges such as deploying the technology demonstrated here in the awake behaving animal where eye motion will be larger than in the anesthetized animal. Nonetheless, the single cone tracking and stimulation experiments in the awake human inspire confidence that this will be possible (*Schmidt et al., 2018a*; *Schmidt et al., 2018b*; *Tuten et al., 2017*). These experiments in which single cones are targeted with light have shown that visibility can be measured from photon absorption with flashes of light delivered through AO to single cones from each of the three foveal cone classes (*Williams et al., 1981 Schmidt et al., 2018b*; *Hofer et al., 2005*), encouraging the possibility that optogenetic stimulation of a single RGC might be sufficient to elicit a sensation in an awake behaving monkey, a hypothesis that future experiments could test. Monkeys are remarkably good at psychophysical experiments, with performance rivalling that of human observers (*Matsuno and Fujita, 2009*).

Interpretation of causal experiments through optogenetic stimulation of a single RGC is inevitably complicated by the fact that perceptual experience depends on cell populations and such experiments would necessarily produce unnatural population activity. How the brain would interpret such an unusual pattern of excitation is not entirely clear, and it may be necessary to control the stimulation of multiple nearby RGCs to develop a clear understanding of their function. Additionally, should it prove possible to realize this paradigm, it could provide valuable data to guide optogenetics-based vision restoration therapies. This could eventually offer the opportunity to simultaneously activate multiple classes of RGCs according to their individual response characteristics, leading to a perceptual experience that more closely resembles that generated by the natural environment in the sighted eye.

## Materials and methods

### Animals and animal care

These experiments were performed in two anesthetized *Macaca fascicularis*, one male (M1) and one female (M2). Macaques were housed in an AAALAC-accredited facility and were provided with care by four full-time veterinarians, five veterinary technicians, and an animal behaviorist from the Department of Comparative Medicine. The animal care staff monitored each animal for signs of discomfort at least twice daily. They were caged in pairs and had free access to lab chow and water. Their diet was additionally supplemented with daily treats including dried fruits, fresh fruits, vegetables, and nuts. All animals have daily access to enrichment items such as puzzle feeders and mirrors and are provided with movies and/or music. More novel enrichment items such as treat-filled bags, snow, or forage boxes were provided weekly. The macaques had access to a large play space with swings and perches on a rotating basis. This study was carried out in strict accordance with the Association for Research in Vision and Ophthalmology (ARVO) Statement for the Use of Animals and the recommendations in the Guide for the Care and Use of Laboratory Animals of the National Institutes of Health. The protocol was approved by the University Committee on Animal Resources for the University of Rochester (Ref 100567). One animal had two lesions of the central fovea, which had removed cone input to many of the labeled RGCs. The RGCs deafferented in this manner are easily distinguishable from those with normal input and were avoided for study in this experiment.

### Injections for expression of optogenetic and calcium indicator

Prior to injection, both animals received daily subcutaneous injections of cyclosporine to reduce the possibility of immune reaction to the injection. Blood trough levels were monitored to titrate the dose into the range of 150–200 ng/ml, and these levels were maintained throughout the testing period. The ocular surface was disinfected before injection with 50% diluted Betadine. Coexpression of both an optogenetic actuator (ChrimsonR) and a calcium indicator (GCaMP6s) enabled both stimulation of RGCs and recording of their responses. This was achieved with a single intravitreal injection containing a mixture of two adeno-associated virus (AAV2) based vectors with a ubiquitous CAG promoter, AAV2-CAG-tdTomato-ChrimsonR and AAV2-CAG-GCaMP6s. These vectors were synthesized by the University of Pennsylvania Vector core. The injection was delivered via a 30-gauge needle with a tuberculin syringe into the middle of the vitreous roughly 2 mm behind the limbus. The injection was made in the right eye of M1 and the left eye of M2 and had a total volume of 75 µl. Post-injection, the eyes were monitored with conventional SLO imaging (Heidelberg Spectralis) for adverse events such as inflammation, which was not present in either animal throughout data collection. As previously demonstrated (*Dalkara et al., 2013*), expression was largely confined to an annulus spanning from 0.5° to 2°. The inner edge of this annular region corresponds to the most centrally located RGCs that receive input from the most central fovea, while the outer edge corresponds to the most eccentric location where we presume the inner limiting membrane remained thin enough for penetration of the viral vector. All RGCs that obtained expression are close enough to the foveal center to be displaced from their receptive fields, allowing for direct optogenetic activation of cell somas without stimulation of the cones that drive them.

## AO imaging
### Anesthesia and animal preparation
Anesthesia and animal preparation were performed by licensed veterinary technicians. Macaques were fasted overnight prior to anesthesia in preparation for ventilation. The morning of an imaging session, macaques were moved to the laboratory and anesthetized with isoflurane. They were placed prone upon a stereotaxic cart, supported by cushions and covered with a warming blanket. The animals were paralyzed for the duration of imaging via an injection of vecuronium bromide. For the duration of paralysis, the animal's heart and respiratory rates, blood oxygenation levels, and electro-cardiogram were monitored by a veterinary technician. The animal's pupils were dilated with drops of phenylephrine chloride (2.5%) and tropicamide (1%), and the eye fitted with a rigid gas permeable contact lens to maintain corneal hydration. The animal's head was stabilized with ear bars and a chin rest, and the animal's whole body could be rotated via the stereotaxic cart to position the animal for imaging. The macaque was monitored for several hours by a veterinary technician after each session to ensure full and safe recovery from the anesthetic drugs. This procedure was repeated no more than once per week on an individual macaque.

### Imaging procedure
High-resolution imaging was performed with an adaptive optics scanning light ophthalmoscope (AOSLO), a complete description of which can be found in *Godat et al., 2022*. A Shack–Hartmann wavefront sensor in conjunction with an 847 nm laser diode source (QPhotonics) was used for wave-front sensing. The wavefront sensor data controlled a deformable mirror (ALPAO) in a closed-loop configuration operating at 10 Hz to provide correction of ocular aberrations. Reflectance and fluorescence signals were collected with photomultiplier tubes. The RGC layer was imaged using the fluorescent light emitted by the calcium indicator excited with light from a 488 nm laser (Qioptiq), and the optogenetic was excited using the 640 nm line of a multi-line laser (Toptica). The cone mosaic was simultaneously imaged with light from a 796 nm superluminescent diode (Superlum) to provide a higher SNR (signal-to-noise ratio) helpful for both real-time stimulus stabilization and post-recording image registration. Both uses are based upon cross-correlation of small strips of the image (*Yang et al., 2014*). For stimulus stabilization, only strips located near the stimulus' intended location were used in the cross-correlation, allowing the stimulus to be repeatedly presented to the same retinal location. This stabilization is necessary because the heartbeat and respiration of

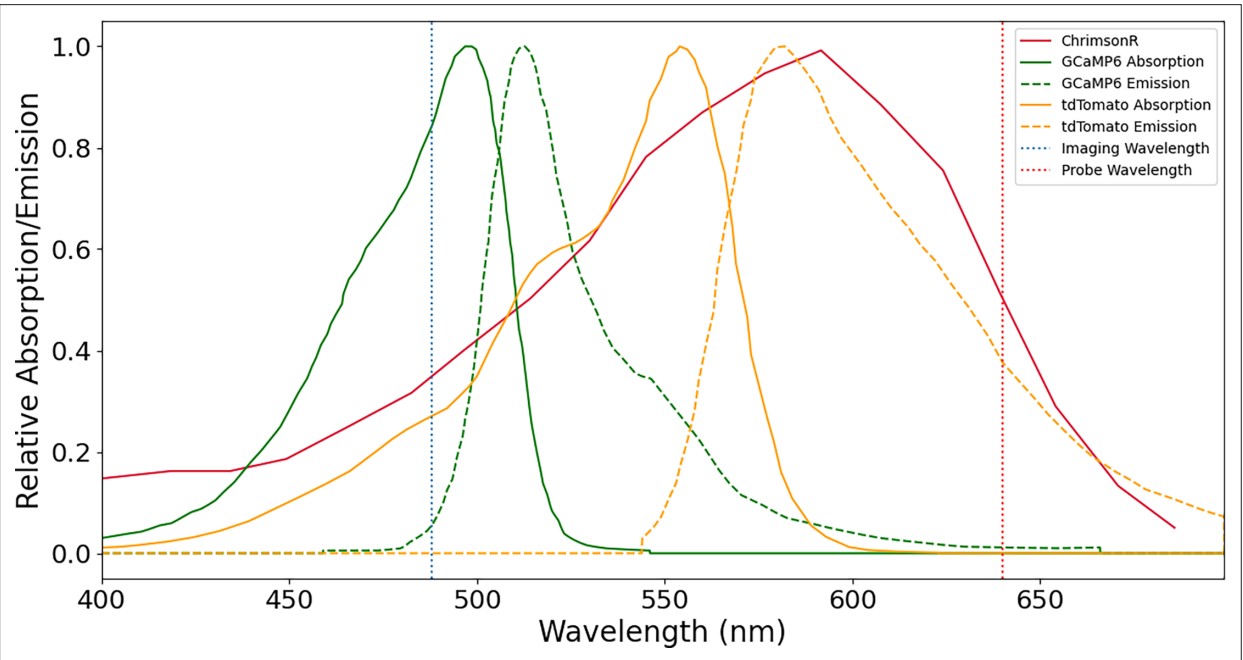

**Figure 3.** Relevant absorption/emission spectra for our fluorophores and optogenetic actuator. The emission wavelengths of tdTomato and GCaMP6 are well suited to activating ChrimsonR, and the wavelength used to image GCaMP fluorescence stimulates both fluorophores well.

the animal cause lateral shifts in the retinal imaging throughout recording of approximately 50 μm. The image registration application is similar in execution and is applied in turn to all sections of the image to obtain a single stabilized video. The motion data from the strip registration of the cone reflectance video was used to stabilize the calcium recording. These methods allowed us to obtain cellular-scale recordings from the primate retina in vivo while correcting for the motion inherent to the living eye.

## Cell selection

For this first study, we selected RGCs lying on the innermost edge of the annulus of expression often had large distances between their somas, making them easier to target with optogenetic stimulation without exciting neighboring cells. This crosstalk could potentially enter our measurements both when stimulating and recording. First, scatter from the 640 nm stimulus may illuminate nearby cell somas or underlying fibers of passage. Additionally, these nearby cell somas, their dendritic arbors, and fibers may contaminate the GCaMP recording if they overlap with the cell of interest's soma. We chose two cells in each of the two animals that were > 20 μm from their nearest neighbor to mitigate the possibility of optical crosstalk. We have considered two possibilities by which light sources in the experiment other than the optogenetic probe might have interfered with the RGC response, but calculations confirm that neither would be expected to appreciably influence the probe response (*Figure 3*). First, the 488 nm imaging light is expected to produce an optogenetic activation that is only about 2% of that of the optogenetic probe . Moreover, it illuminates all cells equally throughout the experiment and would be expected to produce a constant activation, unlike the probe which is confined in space and time. Second, the emission from GCaMP and tdTomato is at least 4 orders of magnitude lower than this 488 nm stimulus, making it unlikely that these fluorophores could have caused a measurable cell response.

Cells within a 1° × 1° field of view were imaged at 488 nm (25 μW) for 30 s to allow for adaptation to the maintained calcium imaging light. The RGCs showed some level of baseline activation in response to this. ChrimsonR was then activated by a circular stimulus (640 nm, 750 μW) with a diameter of 12.5 μm, approximately the size of a cell soma. This stimulus was formed by modulating the 640 nm laser source's output to only be on while scanning over the soma of interest at a rate of 25 Hz. The duration of excitation was 800 ms, throughout which the stimulus was stabilized on the cell's soma using the method described in the imaging procedure section. This stimulus was repeated four times for each trial with a waiting period of 15 s between stimuli. We have seen no evidence that this modulated cell activity via the normal cone pathways.

## Analysis

The GCaMP6s recordings were registered using strip registration and the frames averaged to obtain a high SNR image of the RGC mosaic. This registration is necessary because the heartbeat and respiration of the animal cause lateral shifts in the retinal imaging throughout recording of approximately 50 μm. The resulting RGC image was manually segmented to create a binary mask that can be applied to each frame of the video. The mask is used to extract the mean brightness of each cell soma for each frame. These brightness values are then used to calculate $\Delta F/F$ values for each cell soma in response to each stimulus. This calculation was performed for all cells that remain within the field of view for the entirety of the recording. Cells near the edge of the field of view often move in and out of visibility, obscuring any signals that may be present. The mean and standard deviation of $\Delta F/F$ values are calculated for periods during which no stimulus was present using data taken from all labeled RGCs in the subject's retina. These are used to transform the cell responses to stimuli into z-scores.

## Acknowledgements

We thank Amber Walker for providing animal care and anesthesia and Sara Patterson for providing feedback on the manuscript and data analysis. We thank the vector core at the Perelman School of Medicine, University of Pennsylvania and the Genetically-Encoded Neuronal Indicator and Effector (GENIE) Project and the Janelia Research Campus of the Howard Hughes Medical Institute, specifically Vivek Jayaraman, PhD, Douglas S Kim, PhD, Loren L Looger, PhD, and Karel Svoboda, PhD.

# Additional information

## Competing interests

Qiang Yang: Patents with the University of Rochester, Canon Inc and the University of Montana, for image stabilization algorithms: US patent 9,226,656: "Real-time optical and digital image stabilization for adaptive optics scanning ophthal moscopy", US patent 9,406,133: "System and method for real-time image registration", US patent 9,485,383, "Imaging based correction of distortion from a scanner" and US patent 9,454,084, "Light source modulation for a scanning microscope". David R Williams: Patents with the University of Rochester for adaptive optics retinal imaging: US patent 6,199,986 "Rapid, automatic measurement of the eye's wave aberration", US patent 6,264,328 "Wavefront sensor with off-axis illumination" and US patent 6,338,559 "Apparatus and method for improving vision and retinal imaging". The other authors declare that no competing interests exist.

## Funding

| Funder | Grant reference number | Author |
|---|---|---|
| National Eye Institute | R01EY031467 | David R Williams |
| National Eye Institute | P30EY001319 | David R Williams |
| United States Air Force Office of Scientific Research | FA9550-22-1-0167 | David R Williams |

The funders had no role in study design, data collection and interpretation, or the decision to submit the work for publication.

## Author contributions

Peter J Murphy, Conceptualization, Data curation, Formal analysis, Investigation, Methodology, Writing – original draft, Writing – review and editing; Juliette E McGregor, Conceptualization, Data curation, Writing – original draft, Writing – review and editing; Zhengyang Xu, Data curation; Qiang Yang, Software, Writing – review and editing; William Merigan, David R Williams, Conceptualization, Supervision, Funding acquisition, Investigation, Methodology, Writing – original draft, Project administration, Writing – review and editing

## Author ORCIDs

Peter J Murphy (ID) https://orcid.org/0000-0002-9885-7779
Juliette E McGregor (ID) http://orcid.org/0000-0002-8046-028X
Zhengyang Xu (ID) http://orcid.org/0009-0004-9355-8405
William Merigan (ID) http://orcid.org/0009-0005-9411-6120

## Ethics

This study was carried out in strict accordance with the Association for Research in Vision and Ophthalmology (ARVO) Statement for the Use of Animals and the recommendations in the Guide for the Care and Use of Laboratory Animals of the National Institutes of Health. The protocol was approved by the University Committee on Animal Resources for the University of Rochester (Ref # 100567).

Reviewer #1 (Public review): https://doi.org/10.7554/eLife.90050.3.sa1
Reviewer #2 (Public review): https://doi.org/10.7554/eLife.90050.3.sa2
Reviewer #3 (Public review): https://doi.org/10.7554/eLife.90050.3.sa3
Author response https://doi.org/10.7554/eLife.90050.3.sa4

# Additional files

## Supplementary files

MDAR checklist

## Data availability

Data used for this publication have been deposited at https://doi.org/10.17605/OSF.IO/ZGJSV and is publicly available.

The following dataset was generated:

| Author(s) | Year | Dataset title | Dataset URL | Database and Identifier |
| --- | --- | --- | --- | --- |
| Murphy P | 2023 | Optogenetic Stimulation of Single Ganglion Cells in the Living Primate Fovea | https://doi.org/10.17605/OSF.IO/ZGJSV | Open Science Framework, 10.17605/OSF.IO/ZGJSV |

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
