## [Editor Report · eLife Assessment]

This article shows that it is possible to optogenetically activate single retinal ganglion cells in vivo in monkeys. This is an **important** step toward causal tests of the role of specific ganglion cell types in visual perception. The article presents **convincing** evidence for the promise of the approach, but further work will be needed to fully explore its limitations and specificity.

---

## [Referee Report · Reviewer #1 (Public review)]

Summary

This manuscript reports preliminary evidence of successful optogenetic activation of single retinal ganglion cells (RGCs) through the eye of a living monkey using adaptive optics (AO).

Strengths

The eventual goals of this line of research have an enormous potential impact in that they will probe the perceptual impact of activating single RGCs. While I think more data should be included, the four examples shown look quite convincing.

Weaknesses

While this is undoubtedly a technical achievement and an important step along this group's stated goal to measure the perceptual consequences of single-RGC activations, the presentation lacks the rigor that I would expect from what is really a methods paper. In my view, it is perfectly reasonable to publish the details of a method before it has yielded any new biological insights, but in those publications, there is a higher burden to report the methodological details, full data sets, calibrations, and limitations of the method. There is considerable room for improvement in reporting those aspects. Specifically, more raw data should be shown for activations of neighboring RGCs to pinpoint the actual resolution of the technique, and more than two cells (one from each field of view) should be tested. Some information about the density of labeled RGCs in these animals would also be helpful to provide context for how many well-isolated target cells exist per animal.

---

## [Referee Report · Reviewer #2 (Public review)]

Murphy et al. expressed ChrimsonR and GCaMP6s in retinal ganglion cells of a living macaque. They recorded calcium responses and stimulated individual cells, optically. Neurons targeted for stimulation were activated strongly whereas neighboring neurons were not.

The ability to record from neuronal populations while simultaneously stimulating a subset in controlled way is a high priority for systems neuroscience, and this has been particularly challenging in primates. This study marks an important milestone in the journey towards this goal.

---

## [Referee Report · Reviewer #3 (Public review)]

This paper reports a considerable technical achievement: the optogenetic activation of single retinal ganglion cells in vivo in monkeys. As clearly specified in the paper, this is an important step towards causal tests of the role of specific ganglion cell types in visual perception. The paper is brief, and it will be important to follow this work with a more detailed methodological description to guide related work, to explore limitations, and to build confidence in the specificity of the approach.

---

## [Author Response]

The following is the authors’ response to the original reviews.

**Reviewer #1 (Public Review):**
SummaryThis manuscript reports preliminary evidence of successful optogenetic activation of single retinal ganglion cells (RGCs) through the eye of a living monkey using adaptive optics (AO).StrengthsThe eventual goals of this line of research have enormous potential impact in that they will probe the perceptual impact of activating single RGCs. While I think more data should be included, the four examples shown look quite convincing.WeaknessesWhile this is undoubtedly a technical achievement and an important step along this group's stated goal to measure the perceptual consequences of single-RGC activations, the presentation lacks the rigor that I would expect from what is really a methods paper. In my view, it is perfectly reasonable to publish the details of a method before it has yielded any new biological insights, but in those publications, there is a higher burden to report the methodological details, full data sets, calibrations, and limitations of the method. There is considerable room for improvement in reporting those aspects. Specifically, more raw data should be shown for activations of neighboring RGCs to pinpoint the actual resolution of the technique, and more than two cells (one from each field of view) should be tested.

We have expanded sections discussing both the methodology and limitations of this technique via a rewrite of the results and discussion section. The data used in the paper is available online via the link provided in the manuscript. We agree that a more detailed investigation of the strengths and limitations of the approach would have been a laudable goal. However, before returning to more detailed studies, we have shifted our effort to developing the monkey psychophysical performance we need to combine with the single cell stimulation approach described here. In addition, the optogenetic ChrimsonR used in this study is not the best choice for this experiment because of its poor sensitivity. We are currently exploring the use of ChRmine (as described in lines 93-97), which is roughly 2 orders of magnitude more sensitive. We have also been working on methods to improve probe stabilization to reduce tracking errors during eye movements. Once these improvements have been implemented, we will undertake the more detailed studies suggested here. Nonetheless, as a pragmatic matter, we submit that it is valuable to document proof-of-concept with this manuscript.

Some information about the density of labeled RGCs in these animals would also be helpful to provide context for how many well-isolated target cells exist per animal.

We agree. Getting reliable information about labeled cell density would be difficult without detailed histology of the retina, which we are reluctant to do because it would require sacrificing these precious and expensive monkeys from which we continue to get valuable information. We are actively exploring methods to reduce the cell density to make isolation easier including the use of the CAMKII promoter as well as the use of intracranial injections via AAV.retro that would allow calcium indicator expression in the peripheral retina where RGCs form a monolayer. It may be that the rarity of isolated RGCS will not be a fundamental limitation of the approach in the future.

**Reviewer #2 (Public Review):**
This proof-of-principle study lays important groundwork for future studies. Murphy et al. expressed ChrimsonR and GCaMP6s in retinal ganglion cells of a living macaque. They recorded calcium responses and stimulated individual cells, optically. Neurons targeted for stimulation were activated strongly whereas neighboring neurons were not.The ability to record from neuronal populations while simultaneously stimulating a subset in a controlled way is a high priority for systems neuroscience, and this has been particularly challenging in primates. This study marks an important milestone in the journey towards this goal.The ability to detect stimulation of single RGCs was presumably due to the smallness of the light spot and the sparsity of transduction. Can the authors comment on the importance of the latter factor for their results? Is it possible that the stimulation protocol activated neurons nearby the targeted neuron that did not express GCaMP? Is it possible that off-target neurons near the targeted neuron expressed GCaMP, and were activated, but too weakly to produce a detectable GCaMP signal? In general, simply knowing that off-target signals were undetectable is not enough; knowing something about the threshold for the detection of off-target signals under the conditions of this experiment is critical.

We agree with these points. We cannot rule out the possibility that some nearby cells were activated but we could not detect this because they did not express GCaMP. We also do not know whether cells responded but our recording methods were not sufficiently sensitive to detect them. A related limitation is that we do not know of course what the relationship is between the threshold for detection with calcium imaging and what the psychophysical detection threshold would have been an awake behaving monkey. Nonetheless, the data show that we can produce a much larger response in the target cell than in nearby cells whose response we can measure, and we suggest that that is a valuable contribution even if we can’t argue that the isolation is absolute. We’ve acknowledged these important limitations in the revised manuscript in lines 66-77.

Minor comments:Did the lights used to stimulate and record from the retina excite RGCs via the normal lightsensing pathway? Were any such responses recorded? What was their magnitude?

The recording light does activate the normal light-sensing pathway to some extent, although it does not fall upon the RGC receptive fields directly. There was a 30 second adaptation period at the beginning of each trial to minimize the impact of this on the recording of optogeneticallymediated responses, as described in lines 222-224. The optogenetic probe does not appear to significantly excite the cone pathway, and we do not see the expected off-target excitations that would result from this.

The data presented attest to a lack of crosstalk between targeted and neighboring cells. It is therefore surprising that lines 69-72 are dedicated to methods for "reducing the crosstalk problem". More information should be provided regarding the magnitude of this problem under the current protocol/instrumentation and the techniques that were used to circumvent it to obtain the data presented.

The “crosstalk problem” referred to in this quote refers to crosstalk caused by targeting cells at higher eccentricities that are more densely packed, which are not represented in the data. The data presented is limited to the more isolated central RGCs.

Optical crosstalk could be spatial or spectral. Laying out this distinction plainly could help the reader understand the issues quickly. The Methods indicate that cells were chosen on the basis that they were > 20 µm from their nearest (well-labeled) neighbor to mitigate optical crosstalk, but the following sentence is about spectral overlap.

We have added a clearer explanation of what precisely we mean by crosstalk in lines 213-221.

Figure 2 legend: "...even the nearby cell somas do not show significantly elevated response (p >> 0.05, unpaired t-test) than other cells at more distant locations." This sentence does not indicate how some cells were classified as "nearby" whereas others were classified as being "at more distant locations". Perhaps a linear regression would be more appropriate than an unpaired t-test here.

The distinction here between “nearby” and “more distant” is 50 µm. We have clarified this in the figure caption. Performing a linear regression on cell response over distance shows a slight downward trend in two of the four cells shown here, but this trend does not reach the threshold of significance.

Line 56: "These recordings were... acquired earlier in the session where no stimulus was present." More information should be provided regarding the conditions under which this baseline was obtained. I assume that the ChrimsonR-activating light was off and the 488 nmGCaMP excitation light was on, but this was not stated explicitly. Were any other lights on (e.g. room lights or cone-imaging lights)? If there was no spatial component to the baseline measurement, "where" should be "when".

Your assumptions are correct. There was no spatial component to the baseline measurement, and these measurements are explained in more detail in lines 240-243.

Please add a scalebar to Figure 1a to facilitate comparison with Figure 2.

This has been done.

Lines 165-173: Was the 488 nm light static or 10 Hz-modulated? The text indicates that GCaMP was excited with a 488 nm light and data were acquired using a scanning light ophthalmoscope, but line 198 says that "the 488 nm imaging light provides a static stimulus".

The 488nm is effectively modulated at 25 Hz by the scanning action of the system. I believe the 10 Hz modulated you speak of is the closed-loop correction rate of the adaptive optics. The text has been updated in lines 217-219 to clarify this.

A potential application of this technology is for the study of visually guided behavior in awake macaques. This is an exciting prospect. With that in mind, a useful contribution of this report would be a frank discussion of the hurdles that remain for such application (in addition to eye movements, which are already discussed).

Lines 109-130 now offer an expanded discussion of this topic.

**Reviewer #3 (Public Review):**
This paper reports a considerable technical achievement: the optogenetic activation of single retinal ganglion cells in vivo in monkeys. As clearly specified in the paper, this is an important step towards causal tests of the role of specific ganglion cell types in visual perception. Yet this methodological advance is not described currently in sufficient detail to replicate or evaluate. The paper could be improved substantially by including additional methodological details. Some specific suggestions follow.The start of the results needs a paragraph or more to outline how you got to Figure 1. Figure 1 itself lacks scale bars, and it is unclear, for example, that the ganglion cells targeted are in the foveal slope.

The results have been rewritten with additional explanation of methodology and the location of the RGCs has been clarified.

The text mentions the potential difficulties targeting ganglion cells at larger eccentricities where the soma density increases. If this is something that you have tried it would be nice to include some of that data (whether or not selective activation was possible). Related to this point, it would be helpful to include a summary of the ganglion cell density in monkey retina.

This is not something we tried, as we knew that the axial resolution allowed by the monkey’s eye would result in an axial PSF too large to only hit a single cell. The overall ganglion cell density is less relevant than the density of cells expressing ChrimsonR/GCaMP, which we only have limited info about without detailed histology.

Related to the point in the previous paragraph - do you have any experiments in which you systematically moved the stimulation spot away from the target ganglion cell to directly test the dependence of stimulation on distance? This would be a valuable addition to the paper.

We agree that this would have been a valuable addition to the paper, but we are reluctant to do them now. We are implementing an improved method to track the eye and a better optogenetic agent in an entirely new instrument, and we think that future experiments along these lines would be best done when those changes are completed.

The activity in Figure 1 recovers from activation very slowly - much more slowly than the light response of these cells, and much more slowly than the activity elicited in most optogenetic studies. Can you quantify this time course and comment on why it might be so slow?

We attribute the slow recovery to the calcium dynamics of the cell, and this slow recovery time is consistent with calcium responses seen in our lab elicited via the cone pathway. Similar time courses can be seen in Yin (2013) for RGCs excited via their cone inputs.

Traces from non-targeted cells should be shown in Figure 1 along with those of targeted cells.

We have added this as part of Figure 2.